# Protective Effects of Hexarelin and JMV2894 in a Human Neuroblastoma Cell Line Expressing the SOD1-G93A Mutated Protein

**DOI:** 10.3390/ijms24020993

**Published:** 2023-01-04

**Authors:** Ramona Meanti, Martina Licata, Laura Rizzi, Elena Bresciani, Laura Molteni, Silvia Coco, Vittorio Locatelli, Robert J. Omeljaniuk, Antonio Torsello

**Affiliations:** 1School of Medicine and Surgery, University of Milano-Bicocca, 20126 Monza, Italy; 2Department of Biology, Lakehead University, Thunder Bay, ON P7B 5E1, Canada

**Keywords:** growth hormone secretagogues, neurodegenerative disease, amyotrophic lateral sclerosis, oxidative stress, neuroprotection

## Abstract

Amyotrophic lateral sclerosis (ALS) is an incurable motor neuron disease whose etiology remains unresolved; nonetheless, mutations of superoxide dismutase 1 (SOD1) have been associated with several variants of ALS. Currently available pharmacologic interventions are only symptomatic and palliative in effect; therefore, there is a pressing demand for more effective drugs. This study examined potential therapeutic effects of growth hormone secretagogues (GHSs), a large family of synthetic compounds, as possible candidates for the treatment of ALS. Human neuroblastoma cells expressing the SOD1-G93A mutated protein (SH-SY5Y SOD1^G93A^ cells) were incubated for 24 h with H_2_O_2_ (150 µM) in the absence, or presence, of GHS (1 µM), in order to study the protective effect of GHS against increased oxidative stress. The two GHSs examined in this study, hexarelin and JMV2894, protected cells from H_2_O_2_-induced cytotoxicity by activating molecules that regulate apoptosis and promote cell survival processes. These findings suggest the possibility of developing new GHS-based anti-oxidant and neuroprotective drugs with improved therapeutic potential. Further investigations are required for the following: (i) to clarify GHS molecular mechanisms of action, and (ii) to envisage the development of new GHSs that may be useful in ALS therapy.

## 1. Introduction

Amyotrophic Lateral Sclerosis (ALS) is an adult neurodegenerative disease characterized by the rapid loss of upper and lower motor neurons in the brainstem, motor cortex and spinal cord. Clinically, ALS patients present progressive and irreversible paralysis of all skeletal muscles, muscle hypotrophy and impaired movement, resulting in death from respiratory failure within 3 to 5 years after the onset of symptoms [1]. Although familial (fALS) and sporadic (sALS) forms of ALS have been described, both versions share some common pathophysiological mechanisms, including gene mutations, oxidative stress, glutamate excitotoxicity, mitochondrial dysfunction, protein aggregation, altered axonal transport and inflammation [2]. 

Superoxide dismutase 1 (SOD1) is a ubiquitous homodimeric enzyme responsible for the conversion of superoxide radicals into molecular oxygen and hydrogen peroxide (H_2_O_2_). The SOD1 variants were identified in 20% of fALS and 5% of sALS patients [3]. Among these variants, the replacement of glycine 93 by alanine (SOD1^G93A^) is responsible for a conformational change that leads to a gain-of-function, in which SOD1^G93A^ produces highly toxic reactive radicals. The accumulation of these reactive radicals leads to a decrease in mitochondrial membrane potential, and increased reactive oxygen species (ROS) and free intracellular calcium (Ca^2+^) levels, which results in nuclear and mitochondrial DNA damage and activation of apoptotic processes [4]. Considering that several ALS genes are involved in the DNA damage response (DDR), and that DNA damage and deficiencies in the DDR may play a widespread role in ALS [5], the study of epigenetic markers of ALS is a novel field of interest. Among all markers of DNA damage, an early cellular response to double-strand breaks (DSBs) is γH2AX, the phosphorylated form of the histone variant H2AX. The H2AX variant is activated in response to oxidative stress insult and is up-regulated in SOD1^G93A^ mouse models, specifically in the spinal cord, frontal cortex and striatum [5], and in motor neurons of ALS patients bearing the C9orf72 repeat expansion [6,7]. 

Amyotrophic Lateral Sclerosis is a treatable, but incurable disease; consequently, the development of novel and effective drugs for improved management of patients with ALS remains an outstanding objective. Currently, riluzole and edaravone are approved by the Food and Drug Administration (FDA) and the European Medicines Agency (EMA). Riluzole extends the patient’s lifespan by only 2–3 months, while the precise clinical efficacy of edaravone remains uncertain and requires evidence from large-scale multi-center clinical trials [8]. 

Growth hormone secretagogues (GHSs) are a large family of synthetic and natural compounds endowed with the following: (i) endocrine activities, as demonstrated by their ability to stimulate the release of growth hormone (GH) and insulin-like growth factor 1 (IGF-1); (ii) extra-endocrine activities, including the stimulation of food intake, anticonvulsant and anti-inflammatory effects, and protection of muscle tissue in different pathological conditions [9,10,11,12,13,14,15,16,17].

Among the GHS, hexarelin is a synthetic hexapeptide that exerts cytoprotective effects, both in vivo and in vitro, by acting at the mitochondrial level in cardiac and skeletal muscle, and with important neuroprotective and anti-apoptotic activities [9,11,12,18]. It has been shown that hexarelin modulates the activation of mitogen-activated protein kinases (MAPKs) and phosphoinositide 3-kinase (PI3K)/protein kinase B (Akt) [19,20], and, thereby, could modulate intracellular Ca^2+^ concentrations [13]. Furthermore, hexarelin protects cells in vitro from apoptosis by inhibiting NO synthesis and reactive oxygen species (ROS) release, modulating caspases activity, as well as the expression of proteins belonging to the BCL-2 family [13,14,15,21,22,23].

A peptidomimetic compound, JMV2894, stimulates GH secretion in vivo and in vitro [24,25] and accelerates the recovery of body mass by protecting skeletal muscle from mitochondrial damage [14,16,24]. In fact, JMV2894 is associated with the following: (i) antagonizing cisplatin (CDDP)-induced muscle wasting, preserving intracellular Ca^2+^ homeostasis and preventing mitochondrial damage in skeletal muscles [9,15,17]; (ii) downregulating autophagy-related genes; (iii) stimulating mitochondrial biogenesis and cellular antioxidant defenses [15].

In this in vitro study, we investigated potential protective effects of hexarelin and JMV2894 on human neuroblastoma cells overexpressing the SOD1-G93A mutated protein in order to ascertain whether these two GHSs could be candidates for the development of new drugs for the treatment of ALS.

## 2. Results

### 2.1. Dose-Response Study with H_2_O_2_ on Cell Viability

Mutations in SOD1 are involved in the blockade of H_2_O_2_ reduction, and it has also been shown that excessive H_2_O_2_ concentration causes SOD1 misfolding and toxicity [26]. Although the effects of H_2_O_2_ are largely dependent upon the cell type used, to mimic the massive oxidative stress conditions reported in ALS patients, SH-SY5Y cells overexpressing the normal SOD1 enzyme (WT), or the mutated form (SOD1^G93A^), were treated with increasing concentrations of H_2_O_2_ (50–200 µM) for 24 h.

The H_2_O_2_ caused a dose-dependent reduction in cell viability of both cell lines, which was significant only in SH-SY5Y SOD1^G93A^ (Figure 1). Specifically, 150 µM H_2_O_2_ was the lowest concentration that significantly (*p* < 0.01), and reproducibly, reduced cell viability in SH-SY5Y SOD1^G93A^ cells, compared to the control (Figure 1); consequently, this concentration was used in all subsequent experiments. Interestingly, it has been reported that, in pathological conditions, the intracellular concentrations of H_2_O_2_ rose up to 150 µM [26,27]. 

### 2.2. Dose–Response Study with GHS for the Treatment of SH-SY5Y SOD1^G93A^ Cells

In order to verify that GHS (hexarelin and JMV2894) did not themselves induce toxicity, SH-SY5Y SOD1^G93A^ cells were treated for 24 h with increasing concentrations of GHS (10 nM-10 µM). Hexarelin and JMV2894 did not affect cell replication (Figure 2A,B); therefore, the 1 µM concentrations of GHS were used in all subsequent experiments.

### 2.3. Effects of GHS on Morphological Changes Induced by H_2_O_2_-Treatment in SH-SY5Y SOD1^G93A^ Cells

Cells were stained, as described in Materials and Methods, and observed with a confocal laser-scanning microscope (LSM 710, ZEISS) in order to characterize morphological changes induced by treatments. Representative photomicrographs for each treatment are shown in Figure 3A. First, we quantified the number of SH-SY5Y SOD1^G93A^ cells in a fixed area by the use of a specific macro for ImageJ software [28]. The number of cells in each field was used for normalizing the data of skeleton analysis, which, in turn, was used to quantify endpoints and process length [12,29,30], since the loss of ramifications is a typical characteristic of morphological cytoskeletal changes in apoptosis. Briefly, the Analyze Skeleton Plugin was applied to skeleton images obtained using ImageJ plugin protocols of original photomicrographs, as described in Material and Methods.

As shown in Figure 3B, 150 µM H_2_O_2_ induced a significant reduction (*p* < 0.001) in the number of SH-SY5Y SOD1^G93A^ cells per field, compared with the control group; interestingly, cell numbers were significantly greater in the group treated with the combination of H_2_O_2_ and 1 µM hexarelin (*p* < 0.001) and JMV2894 (*p* < 0.05), compared to controls. Figure 3C,D shows that H_2_O_2_ treatment alone caused a reduction of both cellular process endpoints and summed process length per cell, compared with controls (*p* < 0.001 and *p* < 0.01, respectively); the effects of H_2_O_2_ alone were significantly inhibited by co-incubation with hexarelin or JMV2894 (Figure 3C: hexarelin: 42.6 ± 3.8%, *p* < 0.001; JMV2894: 43.9 ± 3.2%, *p* < 0.001. Figure 3D: hexarelin: 31.3 ± 5.6%, *p* < 0.05; JMV2894: 33.2 ± 5.2%, *p* < 0.05).

We used FracLac for ImageJ to investigate and quantify morphological changes of cells caused by the treatments.

We quantified, by fractal analysis, the following parameters:1-Fractal dimension (D), an index of cell complexity pattern, that is used to identify cellular forms, ranging from simple rounded to complex branched [12,30,31].2-Lacunarity, a property of the soma, based on the heterogeneity or translational and rotational invariance in a shape; lower lacunarity values indicate a loss of shape heterogeneity [12,30,31].3-Maximum Span Across the Convex Hull (MSACH), which is the maximum distance between two points across the convex hull.4-Perimeter, calculated as the number of pixels on the outlined cell shape.5-Area, quantified as the total number of pixels present in the filled shape of the cell image.

As shown in Figure 4A, H_2_O_2_ induced in SH-SY5Y SOD1^G93A^ cells a significant reduction (*p* < 0.01) of D, compared with controls, indicating a reduced branch complexity, according to skeleton analysis. Conversely, SH-SY5Y SOD1^G93A^ cells treated with the combination of H_2_O_2_ and 1 µM GHS exhibited a significantly greater D value (*p* < 0.05), compared with cells treated with H_2_O_2_ alone, with values similar to the D value of controls.

The lacunarity values of cells treated with 150 µM H_2_O_2_ alone were significantly smaller than those of the control group. Hexarelin and JMV2894 antagonized the effects of H_2_O_2_, since cells treated with hexarelin or JMV2894 and H_2_O_2_ showed values significantly greater (*p* < 0.05) than those of cells treated with H_2_O_2_ alone (Figure 4B). Finally, H_2_O_2_ treatment alone significantly reduced MSACH, perimeter and area (*p* < 0.01, *p* < 0.001 and *p* < 0.05, respectively); this reduction was blunted by co-incubation with hexarelin (Figure 4C–E).

### 2.4. H_2_O_2_-Induced Modulation of BCL-2 Family mRNA Levels in SH-SY5Y WT and SOD1^G93A^ Cells

Mitochondria play a crucial role in cell apoptosis by activating the BCL-2 protein family. Specifically, Bax is a pro-apoptotic molecule responsible for the release of cytochrome C from mitochondria, which causes the subsequent activation of effector caspases. By comparison, Bcl-2 exerts anti-apoptotic activity, favoring cell survival. Thus, the apoptotic mechanism involves inhibition of Bcl-2, compared with activation of Bax.

Figure 5A shows that in SH-SY5Y WT and SOD1^G93A^ cells, H_2_O_2_ stimulated a significant increase of pro-apoptotic Bax mRNA levels in a concentration-dependent manner, while tending to increase the anti-apoptotic Bcl-2 mRNA levels only in SH-SY5Y SOD1^G93A^ (Figure 5B). However, as shown in Figure 5C, the ratio of a pro-apoptotic gene (Bax) to the anti-apoptotic gene (Bcl-2) was significantly increased by treatment with H_2_O_2_ in both cellular lines. Surprisingly, the ratio of Bax/Bcl-2 was lower in the SH-SY5Y SOD^G93A^ cells compared to the SH-SY5Y WT cell line. These results implied to the possibility of other forms of death in addition to apoptosis in SH-SY5Y SOD1^G93A^ cells, like an overactive autophagy or necroptosis pathway.

### 2.5. Effects of GHS on H_2_O_2_-Induced Modulation of BCL-2 Family mRNA Levels in SH-SY5Y SOD1^G93A^ Cells

GHS, particularly hexarelin, potently antagonized the oxidative stress effects of 150 µM H_2_O_2_ in cells incubated for 24 h. The levels of Bax mRNA, which were increased by H_2_O_2_ (*p* < 0.001), were significantly (*p* < 0.05) reduced by co-incubation with hexarelin or JMV2894. However, only hexarelin significantly (*p* < 0.05) increased Bcl-2 mRNA levels (Figure 6A,B). These data suggested that GHS treatment protected cells by inhibiting the pro-apoptotic pathway activated by oxidative stress (Figure 6C). 

### 2.6. Effects of H_2_O_2_ on Caspases-3 and -7 mRNA Expressions

The activation of caspase-3 and caspase-7 by H_2_O_2_ occurs at an early stage of apoptotic cell death; therefore, we characterized caspase-3 and caspase-7 mRNA levels. Our results demonstrated that H_2_O_2_, in a concentration dependent manner, over 24h of treatment, induced a significant upregulation of caspase-3 mRNA expression levels in both SH-SY5Y WT and SH-SY5Y SOD1^G93A^ cells. However, it could be observed that the caspase-3 mRNA levels in the SH-SY5Y SOD1^G93A^ line were much higher than in the WT line. In parallel, an increase in expression of caspase-7 mRNA was observed, but this was only significant in SOD1-mutated cells (Figure 7A,B).

### 2.7. Effects of GHS on Caspases-3 and -7 mRNA Expressions and Protein Levels Induced by H_2_O_2_-Treatment

As shown in Figure 8A, H_2_O_2_ induced an increase of caspase-3 mRNA levels (*p* < 0.001) which was significantly antagonized only by hexarelin (*p* < 0.05), while JMV2894 induced only a trend toward its reduction. Additionally, GHS did not reduce the effects of 150 µM H_2_O_2_ on caspase-7 mRNA levels (Figure 8B).

We also quantified the cellular contents of activated caspase-3 and -7 proteins in SH-SY5Y SOD1^G93A^ cells; both species are effector caspases, which are activated through proteolytic processing by upstream caspases to produce the mature subunit. The protein levels of caspase-3 and caspase-7 measured by Western blot analysis were consistent with their mRNA levels. Hexarelin was the only GHS that antagonized the apoptotic mechanism induced by H_2_O_2_ reducing the levels of cleaved caspase-3 (*p* < 0.01, Figure 9A,B).

### 2.8. Effects of GHS on ERK 1/2, p38 and Akt Protein Levels in H_2_O_2_-Treated Cells

We hypothesized that GHS could also modulate MAPK signaling. Among members of the MAPK family, ERK and p38 are known to be associated with cell death or survival [19].

In SH-SY5Y SOD1^G93A^ cells, H_2_O_2_ induced an increase in *p*-ERK/t-ERK, p-p38/t-p38 and p-Akt/t-Akt ratios (*p* < 0.001), whereas 1 µM hexarelin and JMV2894 alone had no effects (Figure 10). By contrast, hexarelin induced a significant reduction in ERK, p38 and Akt phosphorylation stimulated by H_2_O_2_ (*p* < 0.01 for all, Figure 10A–C). In comparison, JMV2894, significantly inhibited only H_2_O_2_ activation of p38 and Akt (*p* < 0.001 and *p* < 0.01, respectively) (Figure 10B,C).

### 2.9. GHS Activity on γH2AX Phosphorylation

Effects of hexarelin and JMV2894 on intracellular levels of γH2AX (the phosphorylated form of the H2AX histone variant at Ser-139 residue) were also evaluated.

The presence of DNA damage in the nucleus was assessed by immunofluorescence using an anti-γH2AX antibody, and counterstained with DAPI and phalloidin (Figure 11A). These studies revealed that a higher percentage of γH2AX positive cells were present when SH-SY5Y SOD1^G93A^ cells were treated with H_2_O_2_, as a consequence of the excessive oxidative stress (*p* < 0.001). As expected, 1 µM GHS had no effects on the percentage of γH2AX positive cells, compared to the control group. Both hexarelin and JMV2894 significantly reduced the DNA damage induced by oxidative stress: in fact, hexarelin and JMV2894 significantly blunted the γH2AX foci distribution (*p* < 0.001), compared to the H_2_O_2_-treatedgroup, where γH2AX coloration was predominantly diffused (Figure 11B).

## 3. Discussion

Amyotrophic lateral sclerosis (ALS) is an adult-onset and progressive neurodegenerative disease caused by the deterioration of motor neurons within the motor cortex, brain stem, and spinal cord, in which oxidative stress (OS) appears intimately linked to several cellular events that contribute to neuronal degeneration and death [32]. The identification of mutations in genes implicated in mitochondrial functions revealed a clear association between OS and ALS onset, even if they were not the only cause of the pathology.

Superoxide dismutase 1 (SOD1) is an enzyme involved in the scavenging of O_2_^−^ to H_2_O_2_ and O_2_, and in the modulation of cellular respiration, energy metabolism, and post-translational modifications [32,33]. It is known that SOD1-related ALS accounts for about 20% of fALS and 5% of sALS, and more than 170 mutations in this gene have been identified [3]. Among all of the mutations, the G93A (glycine 93 substituted for alanine) mutation is the best studied, and is characterized by the loss of antioxidant capability, due to a gain-of-function which affects the following: (i) the enzyme’s change in affinity to natural and abnormal substrates; (ii) causes the enzyme’s aggregation in neurons to increase [34].

Therefore, the coexistence of OS, defective mitochondrial function, and the SOD1^G93A^ mutation seem to be determinants of neuronal degeneration in ALS and are strictly linked to apoptosis mechanisms [35,36].

In this study, we investigated potential therapeutic effects of growth hormone secretagogues (GHSs) in a simplified in vitro model, using the human neuroblastoma cell line (SH-SY5Y), which expresses the SOD1-G93A mutated protein, and treating these cells with H_2_O_2_ (150 µM) for 24 h, in order to mimic the increased oxidative stress environment typical of ALS [26,27]. In parallel, we also evaluated the effects of H_2_O_2_ in SH-SY5Y WT cells, which overexpress the human healthy SOD1 enzyme.

Among GHSs, we selected hexarelin and JMV2894 for their potential neuroprotective activities, as demonstrated in our previous studies [14,15,24].

Hexarelin, a synthetic hexapeptide ligand of the GHS-R1a receptor, has been shown to stimulate cell proliferation of adult hippocampal progenitors (AHP) and to protect against growth factor deprivation-induced apoptosis and necrosis [37], principally through the activation of PI3K/Akt pathway [18,38]. Interestingly, hexarelin also blunts some inflammatory processes activated by neurodegenerative diseases, stroke, and tumor invasion [39], by modulating the release of pro-inflammatory mediators, such as cytokines, reactive oxygen species, free radical species, and nitric oxide, which could contribute to both neuronal dysfunction and cell death [11,40]. Hexarelin also exerts cardioprotective effects, attenuating cardiomyocyte hypertrophy and apoptosis [41], and attenuating mitochondrial abnormalities reported in cancer cachexia [15,42], stimulating biogenesis, mitochondrial mass, and dynamics restoration, reducing expression of autophagy-related genes and ROS production.

JMV2894 is a synthetic agonist of GHS-R1a, as demonstrated by its ability to stimulate the following: (i) Ca^2+^ mobilization in vitro and (ii) growth hormone release in neonatal rats [14]. Similar to hexarelin, JMV2894 antagonizes cisplatin-induced weight loss in rats, restoring body weight to levels similar to controls, without stimulating perirenal and epididymal fat accumulation, but directly acting on skeletal muscle [15,42]. Interestingly, JMV2894 acts through the following: (i) modulation of mitochondrial biogenesis and function; (ii) increase of fusion index and muscle mass; (iii) changes in the expression of autophagy-related genes (Akt/FoxO pathway); (iv) reduction of ROS production [15].

We demonstrated that 24 h treatment with H_2_O_2_ resulted in a significant and concentration-dependent reduction in SH-SY5Y SOD1^G93A^ cell viability. This loss of survival was mainly correlated with an apoptotic mechanism, whereby H_2_O_2_ upregulated mRNA expression of key apoptotic markers.

Interestingly, the lowest concentration that reproducibly, and significantly, inhibited cell growth in SH-SY5Y SOD1^G93A^ cells was the 150 µM, which has been widely reported as the intracellular concentration observed in cellular models of ALS [26,27,43].

The same treatment in SH-SY5Y WT cells showed that increasing H_2_O_2_ concentrations corresponded to a higher expression of apoptosis markers, but this rise was not as pronounced, as in the mutated SOD1-expressing cell line, and did not correlate with an actual loss of cell number, as demonstrated by the MTT assay. We concluded that, at high concentration, H_2_O_2_ also induced toxicity in the WT cell line, which, however, did not exhibit a complete activation of the programmed cell death pathways, probably due to its increased scavenger activity [4].

However, we found that the SH-SY5Y SOD1^G93A^ cells showed a Bax/Bcl-2 ratio lower than observed in the control cell line. This result seemed to be controversial, and suggested that the apoptosis pathway activated in the SH-SY5Y SOD1^G93A^ cell line was not solely affected. Other mechanisms might also be induced and, thereby, affect the fate of cells such as the following: (i) autophagy, (ii) ferroptosis and/or (iii) necroptosis (in an integrated manner). In fact, abnormalities in autophagy occur in ALS pathogenesis and neurodegeneration, with the generation of toxic fragments of SOD1 and acceleration of muscle atrophy [44]. In addition, ferroptosis, which is triggered by oxidative stress, is strictly correlated with apoptosis, and is associated with mitogen-activated protein kinases (MAPKs), leading to selective motor neuron death in ALS [45]. Finally, necroptosis, frequently associated with caspase-independent mechanisms which mediate inflammatory forms of cell death, has been demonstrated in motor neuron death in different models of ALS [45].By comparison, various concentrations of GHS (hexarelin and JMV2894) did not affect cell viability of SH-SY5Y SOD1^G93A^ cells, and, therefore, we used the 1 µM concentration for all subsequent experiments, consistent with previous experiments [12,14].

Apoptosis is known to be one of the most sensitive biological markers for evaluating oxidative stress caused by an imbalance between ROS generation and efficient activity of antioxidant systems [46,47]. It is an active process initiated by genetic programs and culminating in DNA fragmentation, and is characterized by morphological changes, including the following: (i) cell shrinkage; (ii) formation of membrane-packaged inclusions called apoptotic bodies [48]; (iii) activation of caspases and nucleases; (iv) inactivation of nuclear repair polymerases [49]; (v) condensation of nuclei [50]. Accordingly, we evaluated different parameters of this process.

The study of morphometric parameters by the application of Skeleton and Fractal analyses, allowed us to observe that H_2_O_2_-treatment induced morphological changes that were characteristic of an apoptotic phenotype, including the following: (i) a drastic loss of cell/field, de-ramification, and reduction of process length; (ii) loss of cellular complexity and shape; (iii) reduction of cell size [51]. In SH-SY5Y SOD1^G93A^ cells, both skeleton and fractal analysis suggested, for the first time, that hexarelin, and to a lesser extent JMV2894, rescued the cellular complexity, ramification, dimension, heterogeneity and shape comparable to values observed in the control group.

Our hypothesis was that GHS could reduce the apoptosis mechanism by the modulation of the intracellular pro-apoptotic signaling molecules belonging to the BCL-2 family. The BCL-2 family consists of two groups of mediators, both of which play important roles in mitochondrial-related apoptosis pathways: (i) the anti-apoptotic group, mainly represented by Bcl-2, and (ii) a pro-apoptotic group, including Bax [52].

Consequently, we quantified, by means of RT–PCR, the effects of GHS on Bax and Bcl-2 mRNA levels. The H_2_O_2_ treatment induced a concentration-dependent activation of pro-apoptotic Bax, and an increase in the Bax/Bcl-2 ratio, confirming the pro-apoptotic effect of the treatment. The GHS treatment alone did not affect mRNA levels of apoptotic signaling molecules compared to the control group, demonstrating that these compounds did not stimulate the apoptosis pathway. At the same time, the reduction in Bax mRNA levels, and the increase in Bcl-2 mRNA levels in the group incubated with the combination of hexarelin and H_2_O_2_ confirmed the potential anti-apoptotic effect of this compound [12,41,53]. Both GHS significantly decreased the Bax/Bcl-2 ratio, demonstrating their efficacy in this in vitro model of ALS.

Proteolytic activation of effector caspases is the key irreversible mechanism that induces cell death by apoptosis [51]. It was, therefore, hypothesized that GHS might exert a protective effect on H_2_O_2_-induced apoptosis by inhibiting the activation of effector caspases [12]. SH-SY5Y SOD1^G93A^ cells treated for 24 h with increasing concentrations of H_2_O_2_ showed significant activation of caspase-3 and -7. Only the treatment with 1 µM hexarelin significantly reduced the activation of caspase-3, both in terms of mRNA levels and protein activation, confirming its anti-apoptotic effects.

To investigate molecular pathways involved in GHS neuroprotection, we quantified the expression of MAPKs (ERK and p38) and PI3K/Akt. MAPKs activation contributes to neuronal dysfunction and is involved in NDDs [54,55]. Furthermore, ERK has been shown to participate in the regulation of cell growth and differentiation, and responses to cellular stress [56]. PI3K/Akt is a key apoptotic modulator in the growth factor signaling pathway [57]. In particular, the phosphorylation of Thr-308 and Ser-473 of Akt serves a key role in modulating the actions of growth factors on cells and plays an important role in neuronal protection [58,59].

Alone, H_2_O_2_ significantly increased the phosphorylation of both MAPKs (ERK and p38) and Akt; by comparison, GHS treatments *per se* (hexarelin and JMV2894) did not alter the p-ERK/t-ERK, p-p38/t-p38 and p-Akt/t-Akt ratio, compared to controls. In sharp contrast, in cells treated for 24 h with the association of GHS and H_2_O_2_, hexarelin significantly reduced the activation of all proteins, while JMV2894 inhibited only the phosphorylation of p38 and Akt.

Interestingly, cumulative evidence supports the involvement of DNA damage due to impaired DNA repair mechanisms in several neurodegenerative disorders, including ALS. Among all epigenetic signals, phosphorylation of the Ser-139 residue of the histone variant H2AX (γH2AX) was upregulated in SOD1^G93A^ mouse models, specifically in the spinal cord, frontal cortex and striatum [5], and in motor neurons of ALS patients bearing the C9orf72 repeat expansion, in comparison to normal subjects [6]. Moreover, H2AX phosphorylation is supposed to be regulated by the caspase-3 pathway and by the MAPK family, especially the p38 protein [60,61].

Our preliminary results, obtained by immunofluorescence visualization of nuclear foci formed as a result of H2AX phosphorylation, demonstrated that hexarelin and JMV2894 significantly decreased the percentage of γH2AX positive cells compared to the H_2_O_2_ treated group. These results suggested a possible involvement of epigenetic modifications in the protective effect of GHS against oxidative injury [62].

Taken together these results suggest that hexarelin and JMV2894 inhibit apoptotic mechanisms induced by H_2_O_2_-treatment in cell lines, through the inhibition of apoptosis and potentiation of MAPKs and PI3K/Akt survival pathways. Further investigations are required to clarify GHS molecular mechanisms of action, and to envisage the development of new GHSs that may be useful in ALS therapy.

## 4. Materials and Methods

### 4.1. Chemicals

Hexarelin, Dulbecco’s Modified Eagle’s Medium/Nutrient Mixture F-12 Ham (DME/F-12), G418 disulfate salt solution, hydrogen peroxide (H_2_O_2_), 3 (4,5 dimethylthiazol-2yl)-2,5-diphenyl tetrazolium bromide (MTT), poly-D-lysine hydrobromide, 4′,6-diamidino-2-phenylindole dihydrochloride (DAPI), fluoromount aqueous mounting medium, and bovine serum albumin (BSA) were purchased from Sigma-Aldrich (St. Louis, MO, USA). Penicillin, streptomycin, L-glutamine, trypsin-EDTA, phosphate-buffer saline (PBS) and fetal bovine serum (FBS) were obtained from Euroclone (Pero, Milan, Italy). Alexa Fluor 488 Phalloidin was purchased from ThermoFisher Scientific (Waltham, MA, USA).

The JMV2894 was synthetized by conventional solid phase from the laboratory of Professor Jean-Alain Fehrentz, Institut des Biolécules Max Mousseron, University of Montpellier (France). Compounds were purified by high performance liquid chromatography (HPLC) (purity ≥ 98%).

Prior to assay, the GHSs were freshly dissolved in ultrapure water. Both GHS and H_2_O_2_ were diluted in culture media to final working concentrations for the experiments.

### 4.2. Cell Culture

Human SH-SY5Y neuroblastoma cells either overexpressing the wild-type (WT) or the G93A mutated SOD1 (SOD1^G93A^), were a kind gift from Professor Lucio Tremolizzo of the University of Milano-Bicocca. Briefly, monoclonal cell lines were obtained by transfection with plasmids directing constitutive expression of either wild-type SOD1 (WT) or mutant G93A (SOD1^G93A^), as described by [4]. The SH-SY5Y cells were grown in DMEM-F12 (Sigma-Aldrich) supplemented with 10% heat-inactivated FBS, 100 IU/mL penicillin, 100 µg/mLstreptomycin (Euroclone) and Mycozap Prophylactic (Lonza), under standard cell culture conditions. SH-SY5Y WT and SOD1^G93A^ cells were grown in presence of 200 µg/mL G418, which was removed 2 days before performing the experiments.

After reaching confluence, SH-SY5Y cells were washed with PBS, detached with trypsin-EDTA solution (Euroclone), and seeded for experiments.

In each experiment, SH-SY5Y SOD1^G93A^ cells were incubated for 24 h with H_2_O_2_ alone or the combination of 150 µM H_2_O_2_ and 1 μM GHS (hexarelin or JMV2894). We have chosen to use these cell models because SH-SY5Y SOD1^G93A^ cells display remarkable biochemical abnormalities compared to the SH-SY5Y WT cells, including the increase of cytosolic Ca^2+^ concentrations, decrease ATP levels, and an increased cytosolic and mitochondrial ROS production [4].

### 4.3. Cell Viability

The SH-SY5Y cells were seeded in 96-well culture plates at the density of 4 × 10^4^ cells/well and cultured for 24 h at 37 °C. The day after seeding, the cells were incubated with increasing concentrations (50–200 µM) of H_2_O_2_ or GHS (10 nM–10 µM). After 24 h of incubation, a 10 µL aliquot of 5 mg/mL MTT (Sigma-Aldrich) was added to each well and incubated at 37 °C for 3 h. Then, the culture medium was removed and a 200 µL aliquot of acidified isopropanol was added in order to dissolve the formazan crystals. Absorbance was read at 570 nm using the multilabel spectrophotometer VICTOR^3^ (Perkin Elmer, MA, USA). Cell viability of control groups was set to 100% and the absorbances of experimental groups were converted to relative percentages (absorbance of experimental group/absorbance of relative control) × 100 = % of viable cells.

### 4.4. Actin Staining Assay

The SH-SY5Y SOD1^G93A^ cells (2 × 10^5^ cells/well) were seeded on coverslips coated with poly-D-lysine (Sigma-Aldrich) and incubated for 24 h. Cells were treated with H_2_O_2_ (150 µM) for 24 h, with, or without, 1 µM GHS (hexarelin or JMV2894), then washed with PBS and fixed with 4% paraformaldehyde (Titolchimica, Rome, Italy) for 10 min at room temperature. Cells were subsequently washed with PBS, incubated with cold acetone for 5 min at −20 °C, and blocked in PBS with 1% BSA for 30 min at room temperature. In order to stain actin, SH-SY5Y SOD1^G93A^ cells were incubated with 2 U/mL Alexa Fluor 488 Phalloidin diluted in PBS with 1% BSA at room temperature for 20 min, and then washed with PBS. Counterstaining of nuclei was made with 1 µg/mL DAPI for 10 min at room temperature. After washing with PBS, fluoromount aqueous mounting medium was added, and the cells were observed under a confocal laser scanning microscope (LSM 710, ZEISS, Jena, Germany); images were captured at 40× and 63× magnification by ZEN software (ZEISS).

### 4.5. Morphological Analysis

Photomicrographs obtained at 40x magnification were used to evaluate the number of cells in the same area using a specifically designed macro with ImageJ software (National Institutes of Health, Bethesda, MD, USA) [12,28]. The same photomicrographs were used for skeleton analysis [12,29]. Skeleton analysis was applied to quantify the number of process endpoints and length normalized by the number of cells in the same area. Briefly, the photomicrographs were filtered to soften the background, enhance the contrast and remove noise, by using the application Fiji free software (https://imagej.net/fiji, accessed on 12 July 2022). All the images were then binarized, subsequently skeletonized and analysed with Analyze Skeleton (2D/3D) plugin (http://imagej.net/AnalyzeSkeleton, accessed on 12 July 2022). All the modification steps and parameters evaluated are illustrated in Figure 12.

Moreover, we applied fractal analysis, using FracLac plugin for ImageJ (https://imagej.nih.gov/ij/plugins/fraclac/fraclac.html, accessed on 12 July 2022), in order to evaluate the cellular shape and morphology by different parameters (fractal dimension, lacunarity, maximum span across the hull, perimeter and area) [12,31]. Photomicrographs obtained with 63x oil immersion objective were modified similarly to skeleton analysis. Photomicrographs of cells were cropped and transformed to 8-bit grayscale images. Then, cell images were binarized and manually edited to obtain a single cell made of continuous set of pixels. To avoid bias, this modification was done taking into account the original image. Binary images were outlined and analyzed with Fractal Analysis plugin. Representative images used for FracLac box counting analysis are shown in Figure 13.

### 4.6. Real-Time PCR Analysis

In order to monitor the apoptosis pathway, SH-SY5Y cells were plated in 24-well culture plates at a density of 2 × 10^5^ cells/well, and treated for 24 h according to previously described protocols. Following treatment, cells were washed with PBS and total RNA was extracted using EuroGOLD Trifast reagent (Euroclone), according to the manufacturer’s instructions, and quantified using a Nanodrop ND-1000 spectrophotometer (Thermo Fisher Scientific, Waltham, MA, USA). Reverse transcription was performed using an iScript cDNA Synthesis Kit (Bio-Rad, Hercules, CA, USA) using 140 ng of RNA for each sample. Amplification of cDNA (21 ng) was performed in a total volume of 20 μL of iTaq Universal Probes Supermix (Bio-Rad), using Real-Time QuantStudio7 Flex (Thermo Fisher Scientific). After 2 min at 50 °C and 10 min at 94.5 °C, 40 PCR cycles were performed using the following conditions: 15 s at 95 °C and 1 min at 60 °C. Relative mRNA concentrations of the target genes were normalized to the corresponding β-actin internal control and calculated using the 2^−ΔΔCt^ method.

### 4.7. Western Blot Analysis

SH-SY5Y SOD1^G93A^ cells were plated in 6-well culture plates at a density of 8 × 10^5^ cells/well, incubated at 37 °C for 24 h and then treated as previously described. Following treatment, cells were rinsed with ice-cold PBS and lysed in RIPA buffer (Cell Signaling Technology, Danvers, MA, USA), supplemented with a protease-inhibitor cocktail (Sigma-Aldrich), according to the manufacturer’s protocol. Total protein concentrations were determined using the Pierce BCA Protein Assay Kit (Thermo Fisher Scientific). Equal amounts of protein (20 μg) were heated at 95 °C for 10 min, loaded on precast 4–12% gradient gels (Invitrogen, Waltham, MA, USA), separated by electrophoresis, and transferred to a polyvinylidene difluoride (PVDF) membrane (Thermo Fisher Scientific). Non-specific binding was blocked with 5% dried fat-free milk dissolved in phosphate-buffered saline (PBS) supplemented with 0.1% Tween-20 (PBS-T) for 1 h at room temperature. After 3 washes in PBS-T, membranes were incubated with the primary antibody overnight at 4 °C (Anti-cleaved caspase-3 (Asp175) (5A1E) rabbit antibody, #9664, Cell Signaling Technology, 1:1000; anti-cleaved caspase-7 (Asp198) rabbit antibody, #9491, Cell Signaling Technology, 1:1000; anti-Phospho-p44/42 MAPK (Erk1/2) (Thr202/Tyr204) rabbit antibody, #9101, Cell Signaling Technology, 1:1000; anti-p44/42 MAPK (Erk1/2) rabbit antibody, #4695, Cell Signaling Technology, 1:1000; anti-Phospho-p38 MAPK (Thr180/Tyr182) rabbit antibody, #4511, Cell Signaling Technology, 1:1000; anti-p38 MAPK rabbit antibody, #9212, Cell Signaling Technology, 1:1000; anti-Phospho-Akt rabbit antibody, #4060, Cell Signaling Technology, 1:2000; anti-Akt rabbit antibody, #4685, Cell Signaling Technology, 1:1000; anti-actin rabbit antibody, #A2066, Sigma Aldrich, 1:2500), and then with a peroxidase-coupled goat anti-rabbit IgG (#31460, Thermo Scientific, 1:5000) for 1 h at room temperature. Signals were developed with the extra sensitive chemiluminescent substrate LiteAblot TURBO (Euroclone) and detected with Amersham ImageQuant 800 (GE Healthcare, Buckinghamshire, UK). Image J software was used to quantify protein bands.

### 4.8. The γH2AX Distribution

The SH-SY5Y SOD1^G93A^ cells were grown on poly-D-lysine coated coverslips (2 × 10^5^ cells/well) for 24 h and treated with H_2_O_2_ (150 µM) for 24 h, with or without 1 µM GHS (hexarelin or JMV2894). Cells were then washed with PBS and fixed with 4% paraformaldehyde in PBS (Titolchimica, Rome, Italy) for 10 minutes at room temperature. Cells were permeabilized in acetone for 10 minutes at −20°C and blocked for 40 min with 1% BSA in PBS at room temperature. To quantify the γH2AX nuclear distribution, cells were incubated with primary anti-Phospho-histone H2AX (Ser 139) (20E3) rabbit antibody (#9718, Cell Signaling Technology, 1:400) at 4 °C overnight. Secondary Alexa Fluor 594 conjugate anti-rabbit antibody (#8889, Cell Signaling Technology, 1:1000) were incubated for 1 h at room temperature, then cells were counter-stained with phalloidin and DAPI and mounted. Images were captured at 40x magnification by ZEN software (ZEISS).

Photomicrographs were used to evaluate the number of blue nuclear DAPI staining and red γH2AX signal in the same area, using a specifically designed macro with ImageJ software (National Institutes of Health, Bethesda, MD, USA) [28]. The quantification of γH2AX nuclear distribution was obtained by the calculation of the ratio of γH2AX-positive nuclei over the total number of nuclei within the specified region of interest: (γH2AX-positive nuclei/DAPI-positive nuclei) × 100 = % of γH2AX^+^ cells.

### 4.9. Statistical Analysis

Statistical analysis was performed using the program GraphPad Prism (GraphPad Software, San Diego, CA, USA). Values are expressed as the mean ± standard error of the mean (SEM). Experiments were independently replicated at least three times. One-way ANOVA, followed by Tukey’s *t*-test, was used for multiple comparisons. A *p*-value of less than 0.05 was considered significant.

## Figures and Tables

**Figure 1 ijms-24-00993-f001:**
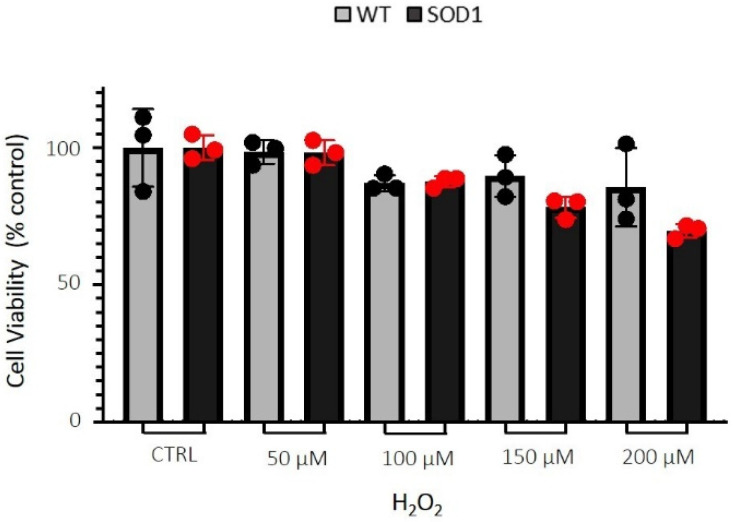
Effects of H_2_O_2_ on the viability of SH-SY5Y WT and SOD1^G93A^ cells. The SH-SY5Y WT and SOD1^G93A^ cells were plated at a density of 40,000 cells/well in 96-well plates. Once confluence was reached, they were treated with increasing concentrations of H_2_O_2_ (50–200 µM) for 24 h. At the end of the treatment, cell viability was assessed by MTT assay. The results are expressed as mean ± SEM of data obtained from 3 independent experiments. In each experiment, groups consisted of 7 replicates for each treatment (total *n* = 21).

**Figure 2 ijms-24-00993-f002:**
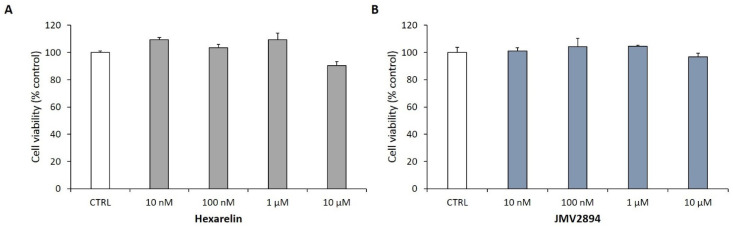
Effects of GHS on the viability of SH-SY5Y SOD1^G93A^ cells. SH-SY5Y SOD1^G93A^ cells were plated at a density of 40,000 cells/well in 96-well plates. Once confluence was reached, they were treated with increasing concentrations of (**A**) hexarelin or (**B**) JMV2894 (10 nM–10 µM) for 24 h. At the end of the treatment, cell viability was assessed by MTT assay. The results are expressed as mean ± SEM of data obtained from 3 independent experiments (*n* = 21).

**Figure 3 ijms-24-00993-f003:**
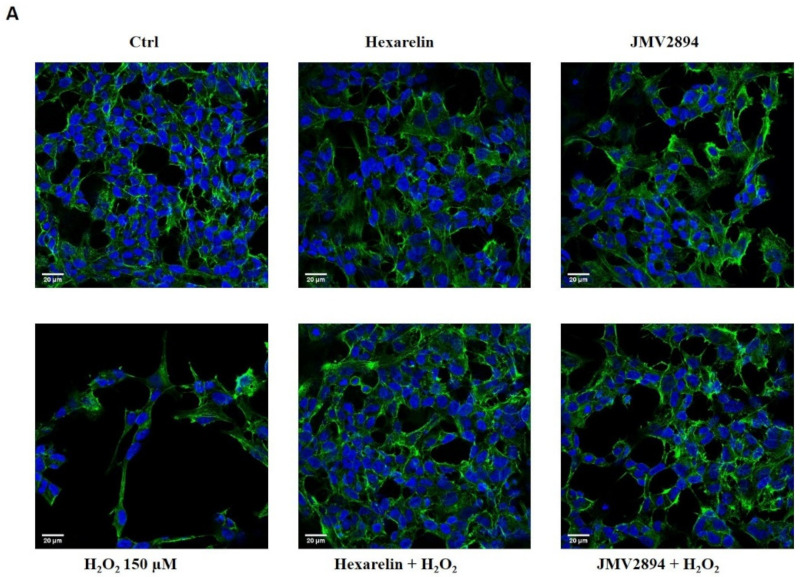
GHS modulated SH-SY5Y SOD1^G93A^ cells de-ramification induced by H_2_O_2_. (**A**) SH-SY5Y SOD1^G93A^ cells were seeded on poly-D-lysine pre-treated coverslips and incubated for 24 h with, or without, GHS and 150 µM H_2_O_2_. At the end of the treatment, cells were fixed and stained for phalloidin and DAPI. Images were captured with a confocal laser scan microscope. Scale bar: 20 µm. Graphical representation of the (**B**) number of cells in the same areas per each treatment; (**C**) number of process endpoints/cells; and (**D**) summed process length/cells. Data are expressed as mean ± SEM replicates obtained in 3 independent experiments (total number of cells analyzed = 100). Statistical significance: ** *p* < 0.01, and *** *p* < 0.001 vs. CTRL; ° *p* < 0.05, and °°° *p* < 0.001 vs. H_2_O_2_.

**Figure 4 ijms-24-00993-f004:**
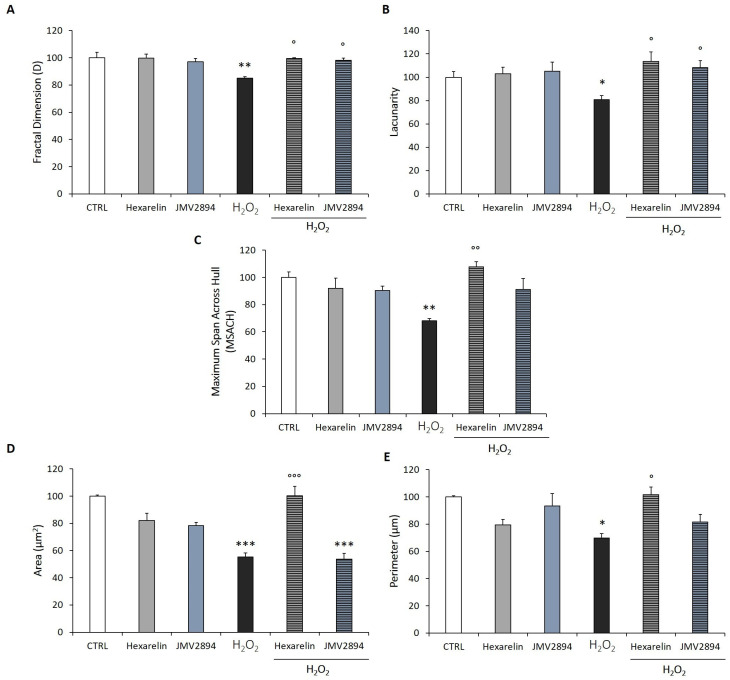
GHS modulation of morphological changes in SH-SY5Y SOD1^G93A^ cells induced by H_2_O_2_ treatment. Numeric representation of: (**A**) Fractal dimension, (**B**) lacunarity, (**C**) maximum span across the convex hull, (**D**) area, and (**E**) perimeter in SH-SY5Y SOD1^G93A^ cells treated for 24 h with 150 µM H_2_O_2_ alone or in combination with GHS. Total number of cells analyzed for each condition = 10. Data are expressed as mean ± SEM. Statistical significance: * *p* < 0.05, ** *p* < 0.01, *** *p* < 0.001 vs. CTRL; ° *p* < 0.05, °° *p* < 0.01, °°° *p* < 0.001 vs. H_2_O_2_.

**Figure 5 ijms-24-00993-f005:**
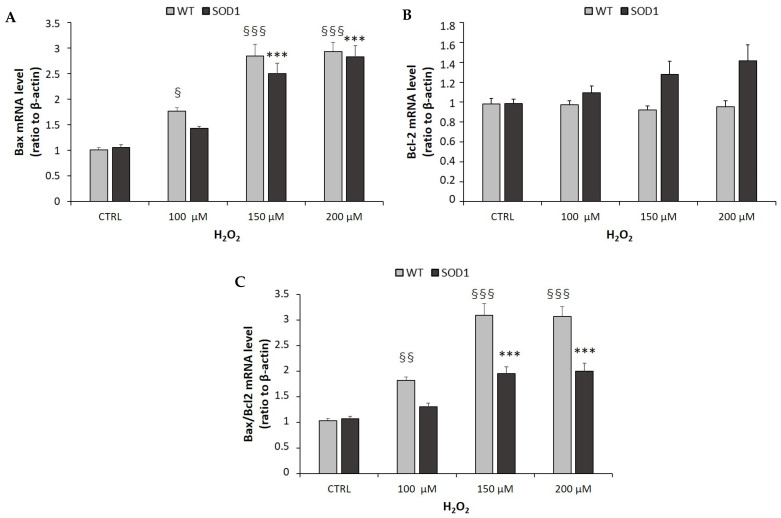
Quantification of mRNA levels of apoptosis markers following H_2_O_2_ exposure. SH-SY5Y WT and SOD1^G93A^ cells were plated at a density of 200,000 cells/well in 24-well plates and treated with different concentrations (0, 100, 150, 200 μM) of H_2_O_2_ alone for 24 h. After treatment, the mRNA expression of Bax (**A**) and Bcl-2 (**B**) was assessed by Real-Time PCR using β-actin as house-keeping gene. (**C**) Quantification of Bax/Bcl-2 ratio. Data are expressed as mean ± SEM of 18 replicates obtained in 3 independent experiments (*n* = 6 in each experiment). For SH-SY5Y WT: ^§^
*p* < 0.05, ^§§^
*p* < 0.01, ^§§§^
*p* < 0.001 vs. CTRL; for SH-SY5Y SOD1^G93A^: *** *p* < 0.001 vs. CTRL.

**Figure 6 ijms-24-00993-f006:**
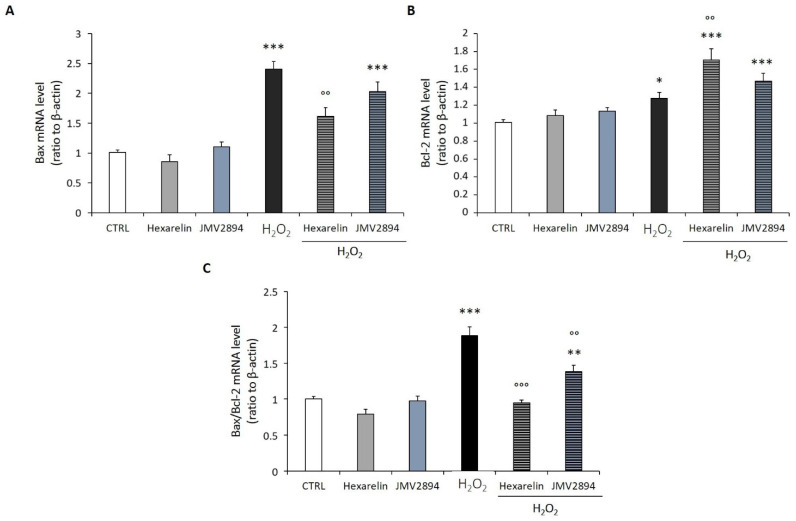
Quantification of mRNA levels of apoptosis markers following H_2_O_2_ and GHS exposure. The SH-SY5Y SOD1^G93A^ cells were plated at a density of 200,000 cells/well in 24-well plates and treated with H_2_O_2_, GHS alone, or co-incubated with GHS 1 µM and H_2_O_2_ 150 µM for 24 h. After treatment, the mRNA expression of Bax (**A**) and Bcl-2 (**B**) was assessed by Real-Time PCR using β-actin as house-keeping gene. (**C**) Quantification of Bax/Bcl-2 ratio. Data are expressed as mean ± SEM of 18 replicates obtained in 3 independent experiments (*n* = 6 in each experiment). * *p* < 0.05, ** *p* < 0.01, *** *p* < 0.001 vs. CTRL; °° *p* < 0.01, °°° *p* < 0.001 vs. H_2_O_2_.

**Figure 7 ijms-24-00993-f007:**
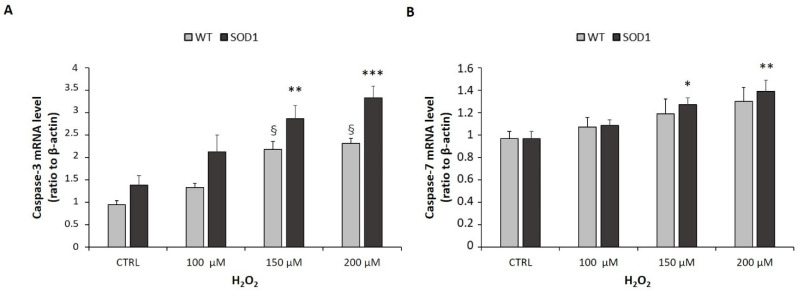
Caspase-3 and caspase-7 mRNA levels following incubation with increasing concentrations of H_2_O_2_. SH-SY5Y WT and SOD1^G93A^ cells were incubated with different concentrations of H_2_O_2_ alone (0, 100, 150, 200 μM). Caspase-3 (**A**) and caspase-7 (**B**) mRNA levels were measured by RT-PCR and normalized for the respective β-actin mRNA levels. Data are expressed as mean ± SEM of 18 replicates obtained in 3 independent experiments (*n* = 6 in each experiment). For SH-SY5Y WT: ^§^
*p* < 0.05 vs. CTRL; for SH-SY5Y SOD1^G93A^ * *p* < 0.05, ** *p* < 0.01, and *** *p* < 0.001 vs. CTRL.

**Figure 8 ijms-24-00993-f008:**
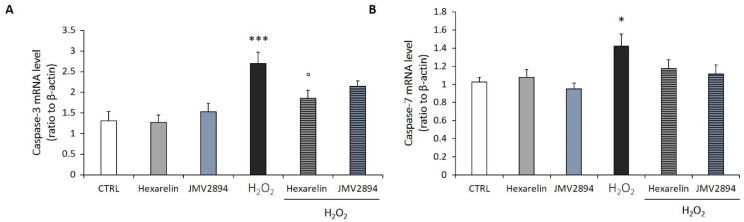
Caspase-3 and caspase-7 mRNA levels following incubation with H_2_O_2_ and GHS. The SH-SY5Y SOD1^G93A^ were also co-incubated with 1 µM GHS and 150 µM H_2_O_2_ for 24 h. Caspase-3 (**A**) and caspase-7 (**B**). The mRNA levels were measured by RT–PCR and normalized for the respective β-actin mRNA levels. Data are expressed as mean ± SEM of 18 replicates obtained in 3 independent experiments (*n* = 6 in each experiment). * *p* < 0.05, and *** *p* < 0.001 vs. CTRL; ° *p* < 0.05 vs. H_2_O_2_.

**Figure 9 ijms-24-00993-f009:**
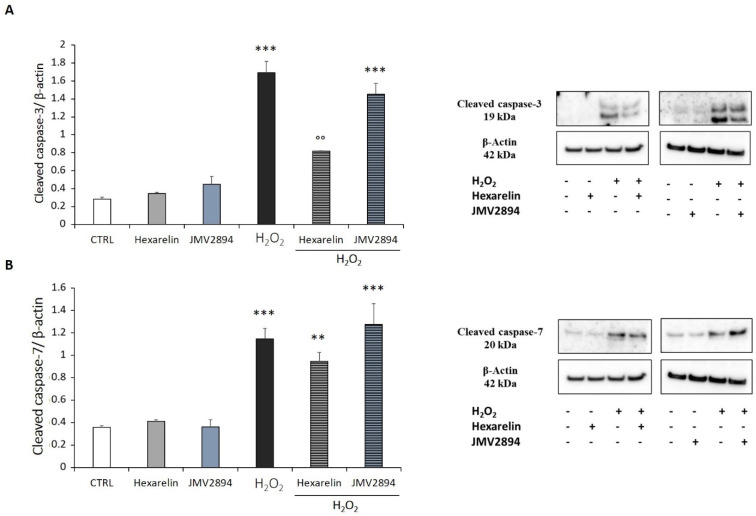
GHS inhibits apoptotic pathway through caspases inactivation. The SH-SY5Y SOD1^G93A^ cells were treated with H_2_O_2_ alone, GHS alone, or with a combination of GHS and H_2_O_2_ for 24 h. Western blot assays were used to measure levels of (**A**) cleaved caspase-3/β-actin, and (**B**) cleaved caspase-7/β-actin. All assays were performed in at least 3 independent experiments (*n* = 3). Statistical significance: ** *p* < 0.01, and *** *p* < 0.001 vs. CTRL; °° *p* < 0.01 vs. H_2_O_2_.

**Figure 10 ijms-24-00993-f010:**
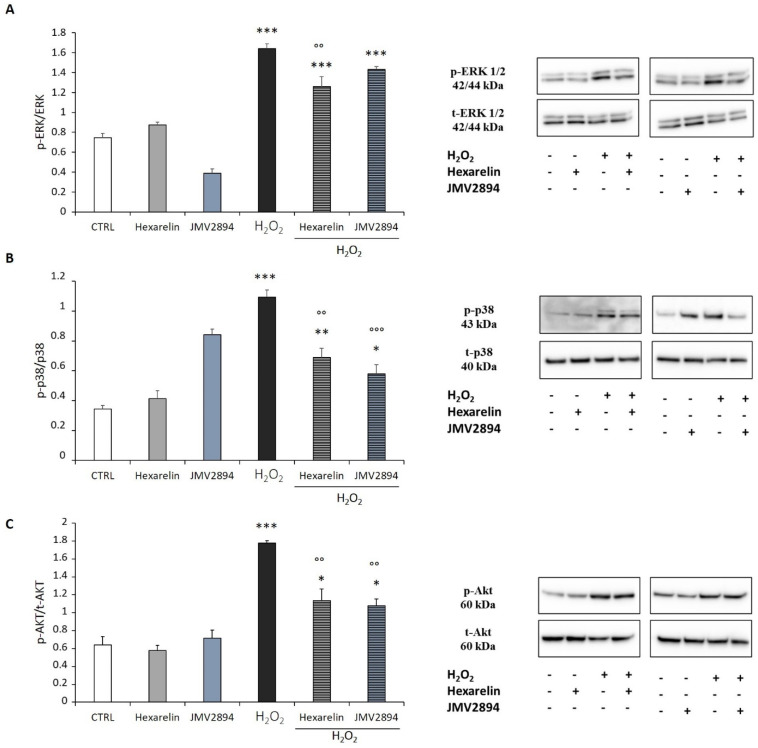
GHS modulate ERK, p38 and Akt activation. SH-SY5Y SOD1^G93A^ cells were treated with or without GHS and H_2_O_2_ for 24 h and western blot assay was used to measure (**A**) p-ERK/t-ERK, (**B**) p-p38/t-p38, and (**C**) p-Akt/t-Akt ratios. All assays were performed in 3 independent experiments. Statistical significance: * *p* < 0.05, ** *p* < 0.01, and *** *p* < 0.001 vs. CTRL; °° *p* < 0.01, and °°° *p* < 0.001 vs. H_2_O_2_.

**Figure 11 ijms-24-00993-f011:**
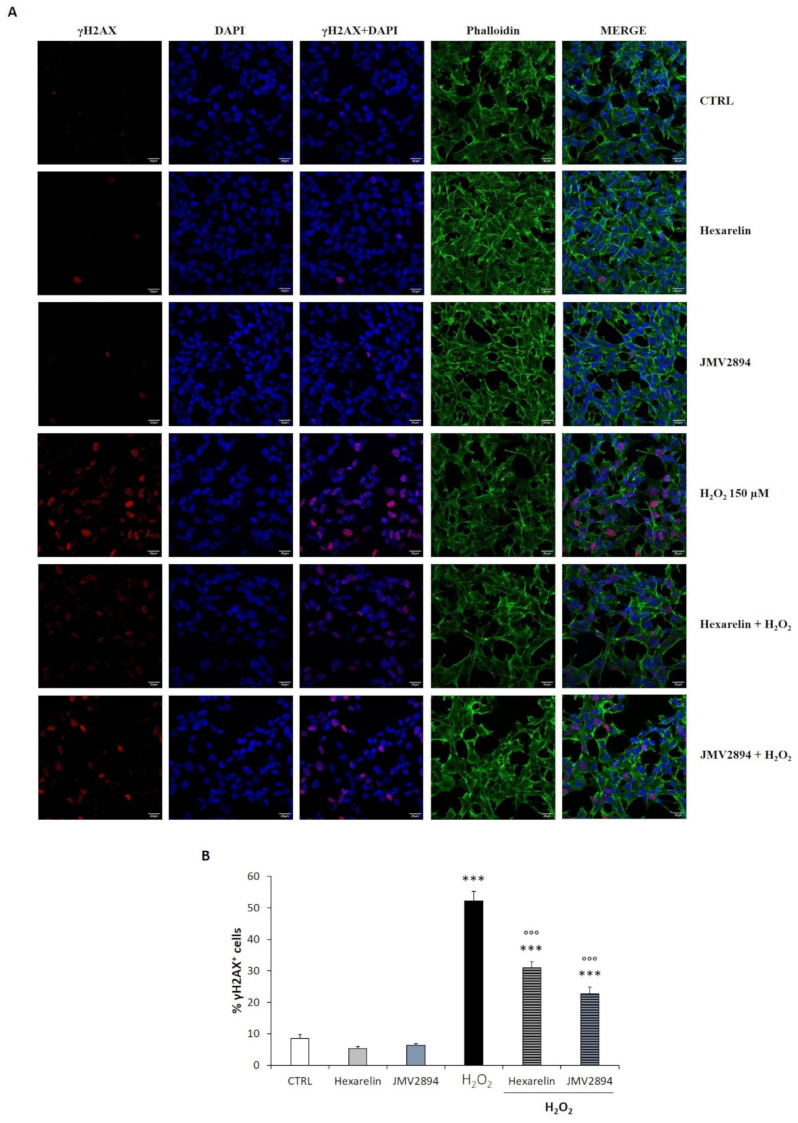
GHS modulated γH2AX foci accumulation in the nucleus induced by H_2_O_2_. (**A**) The SH-SY5Y SOD1^G93A^ cells were seeded on poly-D-lysine pre-treated coverslips and incubated for 24 h with or without GHS and 150 µM H_2_O_2_. At the end of the treatment, cells were fixed, incubated with anti-γH2AX antibody and stained for phalloidin and DAPI. Images were captured with a confocal laser scan microscope. Scale bar: 20 µm. (**B**) Graphical representation of the γH2AX nuclear distribution obtained by the calculation of the ratio of γH2AX-positive nuclei over the total number of nuclei within the specified region of interest. Data are expressed as mean ± SEM replicates obtained in 3 independent experiments (total number of cells analyzed = 100). Statistical significance: *** *p* < 0.001 vs. CTRL; °°° *p* < 0.001 vs. H_2_O_2_.

**Figure 12 ijms-24-00993-f012:**
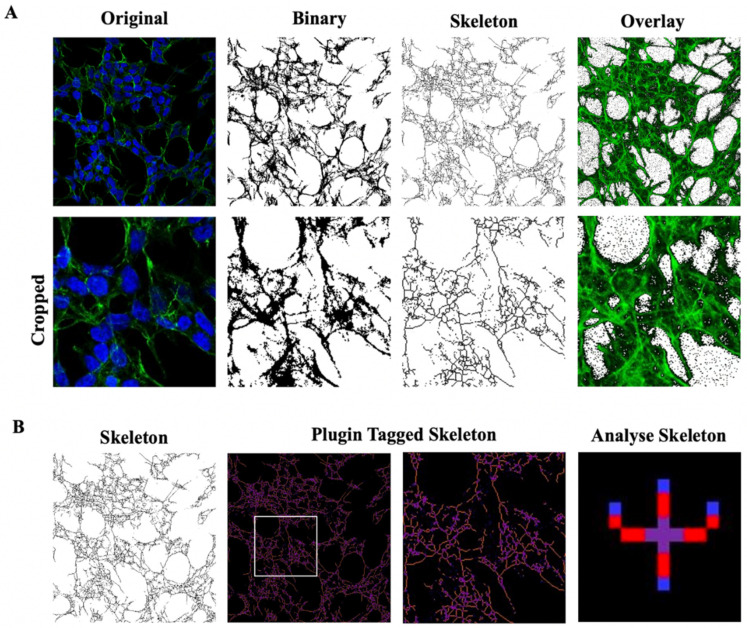
Skeleton Analysis application to quantify cells morphology. (**A**) Representative photomicrograph (40× magnification) and the series of ImageJ plugin protocols of SH-SY5Y cells, which were applied to each photomicrograph for skeleton analysis. Original photomicrograph was modified enhancing the background, removing noise and using with FFT filter prior to be converted to binary images. Binary image was skeletonized. The overlay of skeletonized and original images is reported; cropped photomicrographs show the image details. Scale bar: 20 μm. (**B**) The workflow used to analyze Skeleton plugin: skeletonized process in orange, endpoint in blue, and junction in purple.

**Figure 13 ijms-24-00993-f013:**
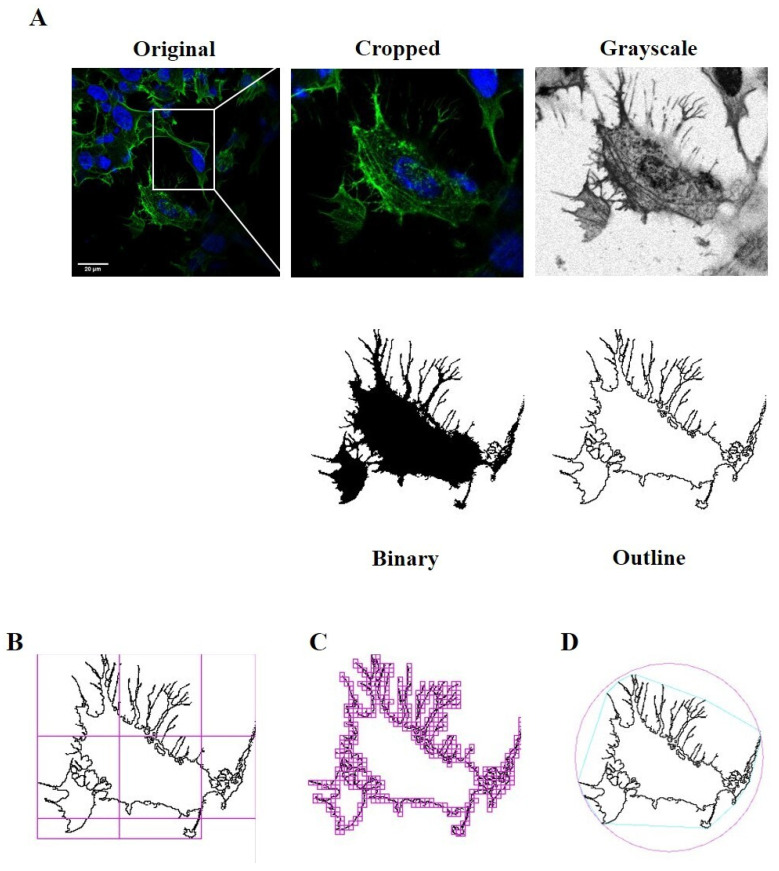
FracLac Analysis application to quantify cellular morphology. (**A**) Representative process applied to obtain an outlined single cell for FracLac plugin. After selecting a cell in the photomicrograph (63× magnification), the image was cropped and modified to remove noise and enhance the background. The image was then processed to obtain an 8-bit grayscale microphotograph, and binarized. Binary image was manually edited to clear the background and to join all branches, and finally outlined. FracLac quantifies cell complexity and shape with a box counting method which permits quantifying fractal dimension and lacunarity (**B**), perimeter and area (**C**), and the maximum span across the convex hull (**D**) by drawing a convex hull (pink) and a bounding circle (green).

## Data Availability

Not applicable.

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
