# Peer review of "Protective Effects of Hexarelin and JMV2894 in a Human Neuroblastoma Cell Line Expressing the SOD1-G93A Mutated Protein"

_ijms, 2023, doi:10.3390/ijms24020993_

Round 1
Reviewer 1 Report
The manuscript submitted by Ramona Meanti and coauthors describes the effects of growth hormone hexarelin on the viability of SH-SY5Y cell line with mutant SOD1 in a model of H2O2-iduced oxidative stress. The work is interesting and can be recommended for publication after some revisions:
1. The authors use 50-200 μM of H2O2 in experiments. Are these doses relevant to real physiological or pathological conditions? If so, please provide appropriate references and discuss it.
2. It would be interesting to see the comparison of H2O2 effects on viability and expression profile of SH-SY5Y and SH-SY5Y SOD1G93A cell lines.
3. To evaluate morphological changes accompanying the apoptosis, the authors calculated some key parameters. It would be better to provide enlarged images of representative cells and indicate graphically (with arrows, lines, etc.) the parameters that were evaluated.
4. Introduction section is well-written and provides all necessary information about the actuality and the importance of the studying issue. However, Discussion section seems as a repeat of experimental conclusions without true discussion and comparison of the obtained data with data of other researchers. Some paragraphs lack the references, while in other paragraphs references support some general theses but not conclusions and speculations of the authors.
Minor point.
-The typographical errors have to be corrected (50-200 M for example).
Author Response
Reply to Reviewer 1:
-The manuscript submitted by Ramona Meanti and coauthors describes the effects of growth hormone hexarelin on the viability of SH-SY5Y cell line with mutant SOD1 in a model of H2O2-induced oxidative stress. The work is interesting and can be recommended for publication after some revisions:
Reply: We thank the Reviewer for his/her kind appreciation of our research
- The authors use 50-200 μM of H2O2 in experiments. Are these doses relevant to real physiological or pathological conditions? If so, please provide appropriate references and discuss it.
Reply: Different studies have demonstrated that the effects of H2O2 largely depend on the cellular model [1,2], and i) very low concentrations of H2O2 (1-10nM) act as a redox molecule, stimulating cell signalling and growth; ii) low concentrations of H2O2 (10-120µM) induce a temporary growth arrest; iii) intermediate concentrations (150-400µM) cause permanent growth arrest; and finally, iv) high concentrations (≥1mM) determine cell necrosis [3,4]. In particular, in ALS it has been demonstrated that intracellular H2O2 concentration increased up to 150µM and that this amount is closely implicated in the misfolding and toxicity of SOD1 in neuronal cells [2,4].
Considering these evidences and the results obtained in the dose/response experiments, we chose to incubate the cell for 24h with 150µM H2O2 in all experiments.
We have now also included a brief justification for this in the manuscript (Lines 91-97).
- It would be interesting to see the comparison of H2O2 effects on viability and expression profile of SH-SY5Y and SH-SY5Y SOD1G93A cell lines.
Reply: We thank the Reviewer for this suggestion. We have added the results of the effects of different H2O2 concentrations on SH-SY5Y cells over-expressing wild-type SOD1 in the graphs in Fig.1, Fig.5A, B and C, and Fig.7A and B.
- To evaluate morphological changes accompanying the apoptosis, the authors calculated some key parameters. It would be better to provide enlarged images of representative cells and indicate graphically (with arrows, lines, etc.) the parameters that were evaluated.
Reply: We think that this is a very good suggestion. Therefore, in order to make it clearer what parameters are taken into account for the assessment of cell morphology, and how the plug-ins have made it possible to quantify these parameters, we have added representative panels in the materials and methods section (Figures 12 and 13).
- Introduction section is well-written and provides all necessary information about the actuality and the importance of the studying issue. However, Discussion section seems as a repeat of experimental conclusions without true discussion and comparison of the obtained data with data of other researchers. Some paragraphs lack the references, while in other paragraphs references support some general theses but not conclusions and speculations of the authors.
Reply: We have amended the Discussion, adding data from the literature that support the results obtained in this study with the respective references. We hope that now this section is more exhaustive.
Minor point.
The typographical errors have to be corrected (50-200 M for example).
Reply: Thank you for pointing out the error, we have corrected the caption in Figure 1.

Reviewer 2 Report
The manuscript by Meanti et al. characterizes the effect of two Growth Hormone Secretagogues (GHS) as potential therapeurics for Amyotrophic Lateral Sclerosis (ALS). This is a neurodegenerative disease for which effective treatments are lacking and thus the development of novel therapeutics is necessary. The authors used an established model for ALS, namely human neuroblastoma cells expressing a mutated form for the SOD-1 gene (G93A), often found in ALS patients. Using several assays that monitor cell survival/morphology, apoptosis, DNA damage, the authors found that the use of these compounds reverse many of the deleterious effects induced upon oxidative stress in the SOD-1G93A background.
In general, the study is interesting and the experiments are well-performed. The manuscript is also clear and well-written.
Specific comments:
The study is mainly based on graphs representing the effect of the two compounds. The authors have included statistics but it is strongly advised to also indicate in the graph the values/points for each independent experiment performed (in a super-plot like version). Also, in many figures is not clear what the n values represents. For example, in Fig. 1, 3 independent experiments, n=21 or Fig. 4, 3 independent experiments, n=18.
There is no comparison of control cells to SOD-1G93A cells for H2O2 sensitivity and the effect of the compounds in control treated cells.
The authors should consider in using at least for a few critical assays another ALS model, such as C9orf72. This should indicate if the compounds could be considered as general ALS potential therapeutics or for specific ALS-mutations such as SOD-1G93A.
The western blot data in Fig. 8 are not convincing, ate least to demonstrate what is quantified in the graph.
Author Response
Reply to Reviewer 2:
- The manuscript by Meanti et al. characterizes the effect of two Growth Hormone Secretagogues (GHS) as potential therapeutics for Amyotrophic Lateral Sclerosis (ALS). This is a neurodegenerative disease for which effective treatments are lacking and thus the development of novel therapeutics is necessary. The authors used an established model for ALS, namely human neuroblastoma cells expressing a mutated form for the SOD-1 gene (G93A), often found in ALS patients. Using several assays that monitor cell survival/morphology, apoptosis, DNA damage, the authors found that the use of these compounds reverse many of the deleterious effects induced upon oxidative stress in the SOD-1G93A background.
In general, the study is interesting and the experiments are well-performed. The manuscript is also clear and well-written.
Reply: We thank the Reviewer for these very kind comments on our research
Specific comments:
- The study is mainly based on graphs representing the effect of the two compounds. The authors have included statistics but it is strongly advised to also indicate in the graph the values/points for each independent experiment performed (in a super-plot like version). Also, in many figures is not clear what the n values represents. For example, in Fig. 1, 3 independent experiments, n=21 or Fig. 4, 3 independent experiments, n=18.
Reply: Thank you for this comment. Following the suggestion, in Figure 1 (panel A) we have now added to each bar the three points that are the media of values obtained in each independent experiment. These points are the media of the 7 values obtained in each experiment for the indicated group. We have used this representation only in panel A, because it helps to understand what has been made, but it is not necessary and could be a little confounding in the other grapshs. In the legend of the figures we explain that bars show the media of 3 independent experiments; in each experiment there were 7 replicates for each treatment group; the final media is obtained from 21 samples for each treatment. We thank the reviewer for pointing out that this could not be clear for the reader.
- There is no comparison of control cells to SOD-1G93A cells for H2O2 sensitivity and the effect of the compounds in control treated cells.
Reply: Thank you for pointing out this problem. We have now included in Fig.1, Fig.5A, B and C, and Fig.7A and B, the results of the effects of H2O2 in SH-SY5Y cells over-expressing wild-type SOD1.
- The authors should consider in using at least for a few critical assays another ALS model, such as C9orf72. This should indicate if the compounds could be considered as general ALS potential therapeutics or for specific ALS-mutations such as SOD-1G93A.
Reply: Thank you for your very important comment. We are aware of the need to test the neuroprotective effects of the GHS also in other in vitro models of ALS (cell lines expressing mutations such as C9orf72 or TDP43). We are unable to run these experiments in the short future, but we are planning to do them in the next months. We have also considered to investigate the effects of GHS in iPSCs obtained from patients.
- The western blot data in Fig. 8 are not convincing, at least to demonstrate what is quantified in the graph.
Reply: Thank you for pointing out this problem. We have replaced the blot in Fig.8 with a different one that we hope could be more representative (Fig. 10C).

Round 2
Reviewer 1 Report
All my comments have been addressed.
Author Response
We thank the reviewer for the comment, and we are glad to have responded to all his/her requests.

Reviewer 2 Report
The revised manuscript is improved and the authors have addressed the majority of my comments.
In fig. 1 the authors show that the used compounds induce higher cell death in the SOD-1 mutant expressing cells compared to control cells. However, in fig. 5, the ratio of Bax/Bcl2 is lower in the SOD-1 mutant expressing cells compared to control. This seems controversial and needs to be addressed, including the possibility of other forms of death in addiiton to apoptosis induction in the SOD-1 mutant cells.
Author Response
We think that this is a very good suggestion. Therefore, in order to make it clearer we have amended the Results (2.4. H2O2-induced modulation of BCL-2 family mRNA levels in SH-SY5Y WT and SOD1G93A cells, lines 214-217) and the Discussion, adding alternative forms of cell death from the literature with the respective references (lines 423-435). We hope that now these sections are more exhaustive.
